# The Rolf Method of Structural Integration and Pelvic Floor Muscle Facilitation: Preliminary Results of a Randomized, Interventional Study

**DOI:** 10.3390/jcm9123981

**Published:** 2020-12-09

**Authors:** Martyna Kasper-Jędrzejewska, Grzegorz Jędrzejewski, Lucyna Ptaszkowska, Kuba Ptaszkowski, Robert Schleip, Tomasz Halski

**Affiliations:** 1Institute of Health Sciences, Opole University, Plac Kopernika 11a, 45-040 Opole, Poland; grzegorz.jedrzejewski@uni.opole.pl (G.J.); lucyna.ptaszkowska@uni.opole.pl (L.P.); tomasz.halski@uni.opole.pl (T.H.); 2Department of Clinical Biomechanics and Physiotherapy in Motor System Disorders, Faculty of Health Science, Wroclaw Medical University, Grunwaldzka 2, 50-355 Wroclaw, Poland; kuba.ptaszkowski@umed.wroc.pl; 3Department of Sport and Health Sciences, Technical University of Munich, Georg Brauchle Ring 60/62, 80992 München, Germany; robert.schleip@tum.de; 4Diploma University of Applied Sciences, Am Hegeberg 2, 37242 Bad Sooden-Allendorf, Germany

**Keywords:** pelvic floor muscles, Structural Integration, surface electromyography

## Abstract

The management of pelvic floor dysfunctions might need to be based on a comprehensive neuro-musculoskeletal therapy such as The Rolf Method of Structural Integration (SI). The aim of the study was to evaluate the pelvic floor muscle (PFM) after the tenth session of SI by using surface electromyography (sEMG). This was a randomized, interventional study. Thirty-three healthy women were randomly assigned to the experimental (SI) or control group. The outcome measures included PFM bioelectrical activity, assessed using sEMG and endovaginal probes. An intervention in the SI group included 60 min of SI once a week, and teaching on how to contract and relax PFMs; in the control group, only the teaching was carried out. In the SI group, a significant difference was found between the PFM sEMG activity during “pre-baseline rest” (*p* < 0.014) and that during “rest after tonic contraction” (*p* = 0.021) in the supine position, as were significant increases in “phasic contraction” in the standing position (*p* = 0.014). In the intergroup comparison, higher PFM sEMG activity after the intervention “phasic contraction” (*p* = 0.037) and “pre-baseline rest” (*p* = 0.028) was observed in the SI group. The SI intervention significantly changes some functional bioelectrical activity of PFMs, providing a basis for further research on a new approach to PFM facilitation, particularly in clinical populations.

## 1. Introduction

Currently, the correct performance of pelvic floor muscle (PFM) contraction and relaxation, in the form of specific training (pelvic floor muscle training, PFMT), is the most important element of the therapeutic process in the conservative treatment of urinary incontinence or pelvic organ prolapse [1,2,3,4,5]. The main goal of PFMT is to facilitate the effective, automatic response of PFM motor units to the increasing pressure in the abdominal cavity in various situations, thus preventing urine leakage [3]. There is, however, a high percentage of women, both healthy and with symptoms of urinary incontinence, who perform incorrect contraction and relaxation of the PFMs. Tibaek et al. showed that 70% of symptomatic women in their study were unable to contract the PFMs, and 97% of them performed this contraction with reduced force [6]. To facilitate the correction of this, isolated PFM activation (contraction) employing endovaginal palpation, biofeedback using endovaginal probes, and verbal instructions [7] is one of the first tasks for pelvic floor physiotherapists before planning PFMsT [8]. According to Bakker et al. [9], it is necessary to establish pelvic floor physiotherapy educational guidelines, based on the principles of evidence-based practice [9]. In this paper, we illustrate the new possibilities for pelvic floor muscle facilitation using a comprehensive neuro-musculoskeletal and holistic intervention such as The Rolf Method of Structural Integration (SI). SI, according to Ida Rolf, is a form of fascial therapy as well as postural and movement re-education. It is focused on balancing body posture, its proper alignment with gravity, and restoring the body’s maximum functional capabilities [10]. The Rolf Method of Structural Integration consists of ten sessions during which the therapist changes the position of individual parts of the body concerning the vertical and gravity field to improve the ergonomics of movements and body posture, and possibly alleviate the reported ailments [11,12]. Each session has its specific course, which typically includes the mobilization of fascial tissue and all major joints [13]. The aims of each session are in Table 1 [12]. The coordinated work of PFMs and other muscles is necessary to maintain the physiological functions of the urinary system [14,15]. A recent study by Zhang et al. [16], performed to characterize the electromyographic activity of the abdominal muscles during micturition in mice, and to evaluate the contribution of the abdominal response to effective urination, proves that such a relationship does exist [16]. Moreover, the normal function of the above-mentioned muscles may be lost in patients with urinary incontinence. For this reason, motor re-education, in the form of learning motor control and postural re-education, including the restoration of normal PFM-related muscle function, should be included in PFMsT [17,18]. Based on the available studies on the restoration of PFM function, it seems that new possibilities, and methods increasing the probability of success in the treatment of pelvic floor dysfunction [19,20,21,22,23], are still being sought. Therefore, the question arises whether a comprehensive form of therapy, such as SI, can facilitate the restoration of the physiological functions of PFMs and serve as another method of their facilitation, supporting the commonly used PFMsT.

## 2. Methods

This was a randomized, interventional study evaluating the bioelectrical activity parameters of PFMs after SI intervention in healthy, young women. The project was designed in accordance with the current guidelines of the CONSORT statement. The study was registered on the international clinical trials platform: doi.org/10.1186/ISRCTN46707309. The study period (recruitment, data collection, and intervention) was January 2019 to March 2020, and the study took place in the Clinical Research Laboratory in the Physiotherapy Department of the Opole Medical School, Poland. All participants gave their informed consent for inclusion before they participated in the study. The study was conducted in accordance with the Declaration of Helsinki, and the protocol was approved by the Ethics Committee of the Opole Medical School, number KB/205/FI2019. The intervention procedure consisted of several stages. Upon enrollment, participants gave written informed consent after a thorough explanation of the procedures involved, and were told that their anonymity would be preserved, and that they could leave the study at any time. Then, the basic data were collected. After that, the participants were randomly assigned to 1 of 2 groups: the SI-intervention group—in which ten sessions of SI were applied—and the control group: without any intervention. Randomization was carried out using computer-generated random numbers (simple randomization). The participants were randomly assigned to the groups in a 1:1 ratio. The following stage involved an assessment of PFM bioelectrical activity with the Glazer Protocol, which includes the series of pre-baseline rest, phasic contractions, tonic contractions, isometric contraction, and post-baseline rest [24]. Then, each participant of the SI group underwent ten sessions of SI intervention (1 session once a week, 60 min per each). After 10 weeks, all participants were given another electromyographic examination. This project was supported by partial financing from a grant within the budget of Opole Medical School, agreement number 63/ROP/CRUZ/2018.

There were 40 women enrolled voluntarily into the study, and they were screened according to the inclusion and exclusion criteria to determine their eligibility for the study. The inclusion criteria were the provision of informed consent to participate in the study, a lack of contraindications for the sEMG measurements, no symptoms of any PFDs (for at least 1 year), and nulliparity. The exclusion criteria were as follows: pregnancy; a history of gynecological surgeries and surgeries within the abdomen, pelvis, or lower limbs in the last 10 years; any neurological symptoms; systemic diseases; lumbar or pelvic pain in the last 6 months; an allergy to nickel; bleeding of any origin; and a body mass index > 28. Seven potential participants were excluded because they did not meet the inclusion criteria or declined to participate. The final study group consisted of 33 healthy women.

A trained SI therapist with four years of practical experience conducted the SI interventions. Each intervention lasted for 60 min, and the patients underwent a total of 10 sessions at the Clinical Research Laboratory in the Physiotherapy Department of the Opole Medical School in Poland. SI therapy was performed in the following order [13] (Table 1).

PFM bioelectrical activity was assessed using the NORAXON EMG and Sensors System (Scottsdale, AZ, USA) TeleMyo DTS with endovaginal probes (Lifecare PR-02, Everyway Medical Instruments Co., Ltd., Taipei, Taiwan). The measurements were conducted before the first SI session. Then, 72 h after the tenth session, the participants were given another sEMG examination. The participants were informed about the procedure and were taught how to correctly perform PFM contractions and relaxation before the sEMG assessment (Table 2). All parameters were measured in the supine and standing position. The measurement procedure was developed based on guidelines from the Surface EMG for Non-Invasive Assessment of Muscles (SENIAM) project and Glazer Protocol [24,25]: “pre-baseline rest” (10 s of PFM activity at rest before functional measurements), “phasic contraction” (10 s measurements, in which participants performed short, quick contractions of the PFM), “tonic contraction” (5 × 10 s contractions, in which the participants tried to contract the PFM and hold for 10 s), “isometric contraction” (in which the participants attempted to hold the PFM contractions for 60 s), and “post-baseline rest” (10 s of the PFM at rest after the functional measurements). The electromyographic signals were subjected to standard post hoc processing. They were rectified and smoothed using the root-mean-square (RMS) algorithm, and to reduce the phase shift, they were subjected to filtering. A narrow band-pass filter with a frequency range of 50 to 1000 Hz was used (finite impulse response filter—FIR filter). The results are presented in microvolts (µV).

Statistical analysis was performed using the Statistica 13 program (TIBCO, Inc., Palo Alto, CA, USA). For measurable variables, arithmetic means, medians, standard deviations, quartiles, and ranges of variability (extreme values) were calculated. The frequency of occurrence (percentage) was calculated for qualitative variables. All quantitative variables were checked with the Shapiro–Wilk test to establish the type of distribution. An intragroup comparison between the results obtained before and after the intervention was performed using the Wilcoxon test. The comparison of the results between the study group and the control group was assessed using the Mann–Whitney U test. The level of α = 0.05 was assumed for all comparisons.

## 3. Results

Forty participants were eligible for the study. Based on the inclusion and exclusion criteria for the study, 33 women took part in the measurements. Figure 1 presents the flow of the participants at each stage of the project. The participants’ demographic and clinical characteristics are shown in Table 3. There were 20 women aged 21–38 years (median = 24 years) in the SI group and 13 women aged 22–32 years (median = 24 years) in the control group. There were no significant intergroup differences in the participants’ characteristics.

In the experimental group, there was a significant difference in the sEMG amplitude of the PFM activity during “pre-baseline rest” and “rest after tonic contraction” (Table 4) in the supine position. In the “pre-baseline rest” after the intervention, the PFM activity was lower by almost 0.6 µV (*p* = 0.014). A similar result was recorded in the measurement of the “rest after tonic contraction”; the PFM activity was lower by 0.5 µV (*p* = 0.021). There was also a significant difference in the sEMG amplitude of the PFM activity during “phasic contractions” in sthe tanding position; after the intervention, the PFM activity was higher by 3 µV (*p* = 0.014) (Table 5). In the other measurements and in the control group (Table 4 and Table 5), no statistically significant differences were noted.

In the intergroup comparison, the PFM activity measured by sEMG after the tenth SI intervention, “phasic contraction” (*p* = 0.037) in the supine position and “pre-baseline rest” (*p* = 0.028) in the standing position, was observed to be higher in amplitude in the SI group than in the control group.

## 4. Discussion

The main aim of this study was to investigate the impact of SI intervention on PFM bioelectrical activity, as a potential new approach to pelvic floor muscle facilitation in healthy, young women. This research is the first to test this specific, comprehensive method of working with the body in terms of its possible use as a method, supporting the known and recommended conservative treatment of pelvic floor dysfunctions. The Pelvic Physiotherapy Education Guideline, published in 2017, contains information facilitating the planning of education in the field of pelvic physiotherapy, emphasizing the important role of evidence-based practice when choosing therapeutic methods [9], and its content has been endorsed by the International Continence Society. Considering the work of physiotherapists, this seems to be a very important aspect—the results of the 2019 research on the treatment of musculoskeletal dysfunctions indicate that most physiotherapists choose therapeutic methods not recommended, or those for which recommendations do not exist at all [28].

The analysis of before and after intervention results in the experimental group showed that ten sessions of SI resulted in a reduction in sEMG activity, noted in the “pre-baseline rest” and “rest after tonic contraction” measurements in the supine position. Similar results regarding the reduction of PFM bioelectrical activity can be seen in Chmielewska et al. [29], where after six weeks of PFM strength training with biofeedback in healthy, continent, nulliparous women, there was a reduction in resting PFM bioelectric activity in both the supine and standing positions. The main causes of urinary incontinence disorders and dysfunctions are being sought in the mechanisms of developing PFM strength, with much less attention being paid to the mechanisms of relaxation, for example, diastolic. This is likely due to the assumption that relaxation is a passive return of the muscle to a resting state after contraction [30,31], and PFMs retain urine due to strong, rapid, and reflex contractions [32]. Recent studies on relaxation indicate, however, that it is a process of active control of muscle deactivation, which is of great importance for their proper functioning [31]. In the International Urogynecological Association (IUGA)/International Continence Society (ICS) joint report [33] from 2017, there is information about the importance of relaxation training, including techniques to decrease sEMG muscle activity or activation, through a variety of methods, including a conscious effort to relax. According to the authors of this report, PFM relaxation can be defined as the ability of muscles to self-control and respond to motor tasks adequately, because the ability to reduce muscle firing is as important for control as the generation of firing. Currently, the PFM relaxation aspect relates to provoked vestibulodynia, impaired voiding or defecation, pelvic pain, sexual dysfunction [34], and post-partum low back and pelvic pain [35], and the role of PFM relaxation for the continence mechanism is still unclear [32]. Hypertonia (increased muscle tension at rest), characteristic of the dysfunctions mentioned above, may contribute to the incorrect sliding of actin and myosin myofilaments, responsible for muscle contraction and relaxation, and may disrupt the generation of muscular strength and endurance [36]. A current, comprehensive review of PFMsT indicates that women with impaired contraction (impaired motor control) are excluded from studies on PFMsT [37], reinforcing the fact that contraction influences PFM force generation. Thus, can the decrease in the bioelectric activity of PFMs translate into the improvement of their functions and restoration of motor control in the case of various types of pelvic floor dysfunction?

The decrease in PFM bioelectric activity observed after SI intervention can be explained by its influence on the central and autonomic nervous systems. In its assumptions, SI is based on the mobilization of the fascia, which, according to Schleip [38,39], is strongly connected with the nervous system thanks to the presence of mechanoreceptors sensitive to touch. In response to their stimulation, through the central nervous system, it causes changes in the tone of some related striated muscle fibers. Since PFMs consist of skeletal muscle fibers, but also smooth muscle [40,41], it appears that the greater potential impact of SI could be directed to the autonomic nervous system, responsible for the stiffness regulation of fascial tissues via the contractile ability of myofibroblasts, which are connective cells with smooth muscle-like contractile properties, altered global muscle tone, and a change in local vasodilatation and tissue viscosity. Additionally, one of the elements of working with the body, according to the SI method, is the so-called pelvic lift, performed during each of the ten sessions. Based on studies by Cottingham et al. [42,43], we know that this technique increased the activity of the parasympathetic nervous system compared to in the control group, in which no such changes were observed. According to the authors, a one-time, three-minute “pelvic lift” technique used in young men triggers the so-called somatovisceral–parasympathetic nervous system reflex, having a relaxing and calming effect on tissue physiology [43].

In 2001, Weiss et al. [44] described the benefits of manual PFM endovaginal and transrectal therapy in patients with overactive bladders and interstitial cystitis (IC). Their research concluded that trigger points in PFMs are not only a source of pain and dysfunction in urination, but also trigger bladder inflammation through the antidromic reflex. Moreover, patients with IC are characterized by hypertonia in PFMs [44]. This study involved 42 patients, of which 10, in addition to assessing the effects of therapy using a subjective questionnaire, underwent electromyographic measurement of PFMs before and after a series of manual therapy. It was observed that the average amplitude of the resting sEMG signal decreased from 9.73 to 3.61 µV, which was a 65% improvement on the pre-therapy values [44]. The results for the electromyographic measurements of that study are comparable to the present results, in which a statistically significant decrease in sEMG amplitude was recorded in the “pre-baseline rest” and “rest after tonic contraction” measurements in the supine position. It seems that the reduction of the bioelectrical activity of PFMs and the reduction of their resting tone are important in cases where increased PFM tone may cause dysfunction and pain [30,45,46].

An updated systematic review of PFMsT in women with urinary incontinence confirms that it is still the most important form of conservative treatment for this type of condition [37]. The assumptions of this project were based on the hypothesis that postural and motor re-education, in the form of specific fascial therapy performed during ten SI sessions, may facilitate the contraction and relaxation of the PFMs, restoring motor control. The significant increase in PFM bioelectric activity during “phasic contraction” observed in both the intra- and intergroup comparisons in the standing position may suggest that the ten SI sessions contributed to an increase in the recruitment of type II (fast-twitch) fibers, which are responsible for fast, forceful contractions. Fast (phasic) contraction velocities play an important role in strength training [32]. These results provide the basis for the team to continue their research in the clinical population. Moreover, we must also bear in mind Ida Rolf’s considerations, which are based on the hypothesis that applying PFMsT to muscles that are imbalanced (e.g., hypertonic) as a result of incorrect body posture will lead to even greater hypertonia and tone disturbances without bringing beneficial, expected therapeutic effects. Therefore, firstly, the balance of the whole body in the field of gravity should be restored, and the pelvis “set” as horizontally as possible [47], in particular. PFMs keep the abdominal and pelvic organs in the correct position, create abdominal pressure (in cooperation with the diaphragm), close the lumen of the urethra, narrow the transverse dimension of the vagina and the urogenital hiatus, and participate in sexual functions and activities, and together with the respiratory diaphragm and abdominal muscles, they perform postural functions [40,48,49,50]. In 2016, Ramin et al. [51] published an extensive anatomical analysis of the so-called “fascial continuum” within the area of the pelvis and torso [51]. If, from a fascial point of view, the abdominal muscles are synergistically connected with the thoracolumbar fascia and the pelvic floor muscles [52], what effect may an incorrect posture have on their functional status and the formation of pelvic floor dysfunctions?

Abnormal body posture is a term commonly used in clinical practice to describe the relationships between different parts of the body in the sagittal, frontal, and horizontal planes that may be considered defective, non-ergonomic, and likely to affect the functioning of internal organs. It is believed that incorrect posture may place an increased strain on the soft tissues and skeletal system, and interfere with the body’s balance [53]. The work of Manshadi et al. [54], studying 160 women with urinary incontinence, showed a statistically significant difference in the asymmetry of the pelvis in a standing position, compared to healthy women. According to the authors, this asymmetry, understood as a lateral shift of the pelvis in the frontal plane, may influence the resting tension of the PFMs [54]. A literature review by Zhoolideh et al. [55], carried out to determine the relationship between the maintained body posture, pelvic structure, muscle tone, and pelvic floor dysfunctions, indicates the need for further research in this area [55].

The influence of pelvic position on the bioelectrical activity of PFMs in menopausal women with urinary incontinence was assessed by Ptaszkowski et al. [56]. The results obtained by the researchers indicate higher resting and functional bioelectrical activity of PFMs both in the posterior pelvic tilt position and during backward movement in the standing position. This seems to be important from the PFMsT perspective, where the starting position causing posterior pelvic tilt and reduction of lumbar lordosis may contribute to better therapeutic effects in women with UI during menopause. Similar results were obtained in studies by Capson et al. [57], where the position of the pelvis in posterior tilt and standing position induced a higher resting bioelectric activity of PFMs [57] in healthy women. Importantly, in terms of the above-mentioned theses of Ida Rolf, only a neutral pelvis position resulted in higher values of bioelectric activity during maximal voluntary contraction or the Valsalva maneuver. This researcher also presents interesting observations about disturbed posture in women with stress urinary incontinence symptoms, where therapeutic intervention, in the form of posture re-education, could contribute to the improvement of tonic and phasic PFM function. The bioelectric activity of PFMs in the standing position was also investigated by Chen et al. [58], who positioned the pelvis in a posterior or anterior tilt due to specific positioning of the ankles and feet [58]. Results differing from those mentioned above were obtained, in that higher bioelectric activity of PFMs was found in women with stress urinary incontinence in the standing position with the pelvis in anterior tilt. Similar results were obtained by Halski et al. [59], in which, in the supine position, a forward inclination of the pelvis generated higher functional and lower resting bioelectric activity of PFMs in menopausal women [42,59].

A 2015 literature review [51] on the anatomical, myofascial relationships between the pelvic floor, abdominal muscles, and back muscles formed an important theoretical basis for this project, referring to the multitude of fascial functions [52]. Based on the characteristic features of fasciae such as the transmission of force, the coordination of movements, stability, and proprioceptive communications throughout the body [52], it seems that any dysfunction along the myofascial chain may cause the overcompensation and dysfunction of other associated muscles [34]. In the case of PFMs, a persistent lack of muscle relaxation and/or heightened muscular activity may lead to involuntary muscle contraction, which in turn can lead to pelvic pain syndrome, involving taut bands and trigger points [60]. These excessively sensitive points, which are painful during compression, may affect the motor functions of the PFMs, reducing their flexibility, strength, and proprioception [61,62]. In her work, Ida Rolf mentions the precursor of PFMsT, referring to the work of Dr. A. Kegel; in his experiment, 65 percent of the women who participated in the study had beneficial results with PFMsT. Ida Rolf, analyzing the disturbed posture in the pelvic area, concluded that perhaps it was the disturbed structure and myofascial imbalance that was the reason for the lack of PFMsT effectiveness in the remaining 35% of the women [47].

## 5. Clinical Implications

Our results and their interpretation, along with other publications, provide us with a further direction for research, focusing on the use of the Rolf Method of Structural Integration concept in certain types of pelvic floor dysfunctions, and the possible inclusion of this comprehensive form of bodywork in recommendations for pelvic floor physiotherapy. It seems that impaired PFM functions and methods for their conservative treatment, due to fascial connections with other regions of the body, should be considered in the aspect of the body as a whole.

## 6. Limitations

The main limitation of this study is the presentation of short-term results with no follow-up (because of the COVID-19 pandemic), and the fact that the participants were healthy. Further studies need to be conducted in larger, more diverse subject populations, and include analysis of fascial and/or muscular features of the lumbar and pelvic regions, as well as postural analysis in three dimensions. In addition, the menstrual cycles of the study participants should be taken into account. Another limitation is that no method other than sEMG was used to evaluate the strength of the PFMs.

## 7. Conclusions

The ten SI sessions contributed to the observed changes in the bioelectric activity of the pelvic floor muscles, especially during the contraction and relaxation phases. This may indicate that SI intervention may be one of the new methods that can be used to facilitate the contraction and relaxation of the pelvic floor muscles. These preliminary results provide the basis for further research towards a new approach to PFM facilitation, especially in clinical populations.

## Figures and Tables

**Figure 1 jcm-09-03981-f001:**
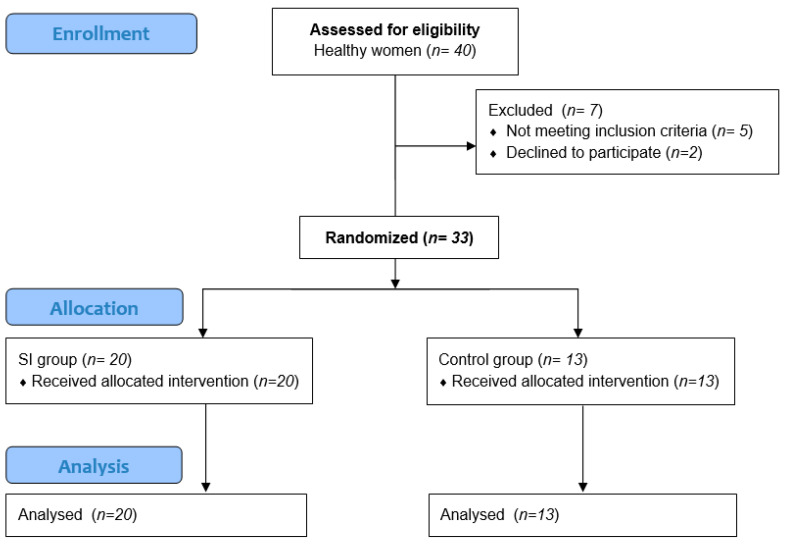
The CONSORT 2010 flow chart of participants in the study.

**Table 1 jcm-09-03981-t001:** Description of ten Structural Integration (SI) interventions [13].

Session	Intervention
1	Increase length and pliability of fascial tissue (FT) on the anterior aspect of the trunk, allowing freer respiratory movement of ribs, and of FT connecting shoulder girdle to rib cage and hips to the pelvis.
2	Increase consistency of FT pliability in feet, ankles, and knees, increasing the support they provide for the upper body.
3	Increase anterior–posterior and cephalic–caudal pliability in FT of the lateral side of the body and left/right and anterior/posterior balance; increase the independence of the thorax from the pelvis.
4	Increase pliability, left/right, and anterior/posterior balance of FT of the medial aspect of legs and pelvic floor.
5	Increase pliability and left/right and surface to deep balance in FT comprising the anterior aspect of the pelvis and lumbar spine.
6	Increase pliability and left/right and surface to deep balance in FT comprising posterior aspect from heel to back.
7	Increase pliability and left/right and anterior/posterior balance in FT of the cranium and cervical spine.
8	Increase fascial tissue pliability and left/right balance in the hands, wrists, elbows, and arms; increase biomechanical flow between upper extremities and spine.
9	Increase fascial tissue pliability comprising the lower extremities through hips and pelvis; increase biomechanical flow between lower extremities and spine.
10	Optimize biomechanical flow through extremities, shoulder, and pelvic girdles to spine; increase overall uniformity of tonus.

**Table 2 jcm-09-03981-t002:** Pelvic floor muscle (PFM) contraction and relaxation protocol.

Participant’s Position	The Therapist
Supine, flexion of hip and knee joints.	Gave verbal instructions regarding the insertion of the endovaginal probe, and the command “contract the muscles around the electrode in your body as much as possible and lift, then completely relax”. These activities were supposed to be an isolated contraction, while the adductors, abdominals, glutes, and extensor muscles of the back were kept relaxed. According to Bø et al. [26], the inability to perform an isolated PFM contraction, for example, without engaging the synergists, may mask the shift in this muscle group and its strength, which in turn translates into lower exercise potential. We recorded the surface electromyography signal when (1) each participant was sure how to contract and relax the PFMs as much as possible without any feedback from the therapist; (2) the sEMG amplitude evaluated was not increased too much, decreased, or absent (an elevated amplitude during resting/baseline activity implies that a muscle has insufficient possibilities for rest); (3) the timing patterns of muscle activity were not too early, late, or asynchronous; (4) the PFM contractions were isolated from synergistic muscles [24]; or (5) the participant stopped breathing or performed a pelvic tilt or straining during the PFM contraction [27]. The sEMG measurement was recorded without eye control (without visualization of sEMG activity on the monitor screen) and with eye control for each participant. As a result, all participants had the same preparation time to achieve muscle control in this region of the body.

**Table 3 jcm-09-03981-t003:** Participants’ characteristics in the experimental (SI) and control groups.

Variables	SI Group (*n* = 20)							Control Group (*n* = 13)							*p*-Value
	$\bar{x}$	Me	Min	Max	Q1	Q3	SD	$\bar{x}$	Me	Min	Max	Q1	Q3	SD	
Age (years)	27	24	21	38	23	29	5	24	23	22	32	23	24	3	0.14 *
Height (cm)	164	164	154	175	159	167	5	159	158	146	174	156	161	7	0.43 *
Weight (kg)	60	61	53	74	54	63	6	57	56	47	63	55	60	5	0.43 *
BMI (kg/m^2^)	21.9	21.3	18.6	29.4	20.6	22.4	2.3	22.6	22.3	20.0	25.4	22.0	23.6	1.4	0.09 *

$\bar{x}$—mean; Me—median; Min—minimum; Max—maximum; Q1—lower quartile; Q3—upper quartile; SD—standard deviation; * Mann–Whitney U.

**Table 4 jcm-09-03981-t004:** Comparison of the sEMG amplitude of PFM (µV) before and after the intervention in the experimental (SI) and control groups in supine position.

sEMG Activity of PFMs (µV)	Measurement	SI Group		Control Group		*p*-Value ^1^
		Me	Q1–Q3	Me	Q1–Q3	
Pre-Baseline Rest	Before	2.5	1.7–3.2	1.5	1.5–2.8	0.98
	After	2.1	1.1–3.2	1.5	1.3–2.0	0.99
*p*-Value ^2^		0.014		0.17		
Phasic Contraction	Before	9.3	6.1–10.0	6.9	4.4–10.2	0.50
	After	9.6	6.9–11.7	6.5	3.2–9.9	0.037
*p*-Value ^2^		0.16		0.13		
Tonic Contraction	Before	9.0	6.3–11.7	6.9	4.0–10.9	0.29
	After	9.4	7.6–12.0	5.5	3.5–13.2	0.15
*p*-Value ^2^		0.25		0.97		
Rest after Tonic Contraction	Before	3.1	2.3–3.9	1.7	1.4–3.4	0.12
	After	2.6	1.9–3.6	1.4	1.2–2.5	0.12
*p*-Value ^2^		0.021		0.10		
Isometric Contraction	Before	8.6	6.4–10.3	5.9	4.8–11.3	0.35
	After	10.0	7.4–12.2	5.9	2.6–8.6	0.06
*p*-Value ^2^		0.33		0.13		
Post-Baseline Rest	Before	2.6	1.8–3.1	1.5	1.4–2.6	0.15
	After	2.3	1.5–2.7	1.5	1.0–2.3	0.14
*p*-Value ^2^		0.18		0.60		

^1^ Mann–Whitney U test; ^2^ Wilcoxon test; sEMG: surface electromyography; PFMs: pelvic floor muscles; µV: microvolts; SI: Structural Integration; Me: median; Q1: first quartile; Q3: third quartile.

**Table 5 jcm-09-03981-t005:** Comparison of the sEMG amplitude of PFM (µV) before and after the intervention in the experimental (SI) and control groups in standing position.

sEMG Activity of PFMs (µV)	Measurement	SI Group		Control Group		*p*-Value ^1^
		Me	Q1–Q3	Me	Q1–Q3	
Pre-Baseline Rest	Before	4.5	3.3–5.8	4.4	3.7–5.3	0.93
	After	5.3	4.2–7.2	4.0	3.9–4.8	0.028
*p*-Value ^2^		0.30		0.75		
Phasic Contraction	Before	8.2	7.2–10.0	10.4	8.8–15.3	0.037
	After	10.7	8.3–16.7	10.2	8.0–14.8	0.78
*p*-Value ^2^		0.014		0.46		
Tonic Contraction	Before	8.6	6.8–12.2	9.2	6.5–13.8	0.41
	After	10.4	7.8–12.8	10.9	6.9–14.2	0.87
*p*-Value ^2^		0.08		0.98		
Rest after Tonic Contraction	Before	4.4	3.5–5.7	4.5	3.7–6.5	0.99
	After	4.7	3.6–6.1	3.8	3.6–4.6	0.21
*p*-Value ^2^		0.97		0.051		
Isometric Contraction	Before	9.0	5.2–12.1	8.2	6.8–11.7	0.63
	After	9.4	6.7–11.7	8.9	8.0–11.3	0.96
*p*-Value ^2^		0.22		0.42		
Post-Baseline Rest	Before	4.0	2.5–6.1	3.8	3.1–5.9	0.84
	After	4.7	3.1–6.0	3.9	2.9–4.2	0.15
*p*-Value ^2^		0.79		0.34		

^1^ Mann–Whitney U test; ^2^ Wilcoxon test; sEMG: surface electromyography; PFMs: pelvic floor muscles; µV: microvolts; IS: Structural Integration; Me: median; Q1: first quartile; Q3: third quartile.

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
