# Peer review of "The Rolf Method of Structural Integration and Pelvic Floor Muscle Facilitation: Preliminary Results of a Randomized, Interventional Study"

_jcm, 2020, doi:10.3390/jcm9123981_

Round 1

Reviewer 1 Report

The search for evidence in Physiotherapy methods is interesting as there are many gaps in the evidence for many of them. However, for there to be evidence we must be rigorous in studies.
In relation to the methodology: Why was the sample size not calculated? 33 women distributed in 2 groups is a small sample.
The inclusion / exclusion criteria do not specify whether the participants were nulliparous. Pregnancies and births produce changes in the abdominal muscles and PFM Only the PFM EMG is performed on the measured variables. This isolated data does not provide sufficient information. What does it mean? What assessment does the ROLF Method make? How do the data collected change after the intervention?
In relation to the results and discussion:
It is necessary to measure in the medium term at least to be able to check if the changes are maintained.
What clinical application do the results have? Is not clear.  

Author Response

Reviewer 1:

Authors’s response: Thank you very much for your review. We answered point by point on your questions and suggestions:

  1. Reviewer comment: In relation to the methodology: Why was the sample size not calculated? 33 women distributed in 2 groups is a small sample.

Thank you for this question. This study will be the basis for calculating the sample size for our further study (registration at ISRCTN: https://doi.org/10.1186/ISRCTN46707309 ). Due to the lack of other reports evaluating The Rolf Method of Structural Integration with sEMG and the lack of our previous research, the sample size was not calculated. It should be emphasized that despite the small size of the groups, some differences can be demonstrated already at this stage of the research. Thank you again for this remark. Information on the calculation of sample size will be added in the next manuscript.

  1. Reviewer comment: The inclusion/exclusion criteria do not specify whether the participants were nulliparous. Pregnancies and births produce changes in the abdominal muscles and PFM.

Authors’s response: Thank you for this suggestion. This is our overlooking during the manuscript preparation. When we planned the project, we assumed that all participants would be nulliparous, especially for the reason you mentioned. We added this information in line: 93.

  1. Reviewer comment: Only the PFM EMG is performed on the measured variables.This isolated data does not provide sufficient information. What does it mean? 

Authors’s response: Thank you for this question. In our study we want to investigate the impact of SI intervention on PFMs bioelectrical activity, as a potential new approach to pelvic floor muscle facilitation in healthy, young women. We began the project by recruiting healthy, young participants, due to the fact that there were no articles testing this specific method of work with the body (The Rolf Method of Structural Integration) particularly in terms of bioelectric activity of the pelvic floor muscles. In the initial phase of the project, we had the opportunity to use only sEMG, so we wanted to present these results as an introduction to the project. Maybe it would be a good idea to supplement the title of our article with the phrase: preliminary results (please see the line 3). The results presented in this manuscript are important because changes in the amplitude values of the recorded sEMG signal steer us in the next stages of the project towards both group of patients with urinary incontinence (and possibly reduced tension and strength of PFM) and chronic pain in the perineum or vulvodynia (hypertonic PFM). It has been reported that elevated amplitude during resting/baseline activity implies that a muscle has insufficient possibilities for rest) (Oleksy et al. 2020).

We know that concluding in the case of sEMG is difficult, but there are still no clear guidelines regarding the interpretation of the results obtained by different authors. On the other hand, there are articles that indicate the diagnostic possibilities of electromyography in the assessment of conservative treatment of urinary incontinence:

- Ptaszkowski K, Malkiewicz B, Zdrojowy R, Paprocka-Borowicz M, Ptaszkowska L. The Prognostic Value of the Surface Electromyographic Assessment of Pelvic Floor Muscles in Women with Stress Urinary Incontinence. J Clin Med. 2020 Jun 23;9(6):1967. doi: 10.3390/jcm9061967. PMID: 32586007; PMCID: PMC7356276.

- Aukee P, Penttinen J, Airaksinen O. The effect of aging on the electromyographic activity of pelvic floor muscles. A comparative study among stress incontinent patients and asymptomatic women. Maturitas. 2003 Apr 25;44(4):253-7. doi: 10.1016/s0378-5122(03)00044-6. PMID: 12697365.

In the next stage of our project we will use e.g. posturography and myoton-pro as a measured variables. We have added a fragment of the above answer to the Limitations (please see the lines 308-309).

  1. Reviewer comment: What assessment does the ROLF Method make? How do the data collected change after the intervention?

Authors’s response:  Thank you for this question. We removed a word “assessment” from the line 48, because this probably misled the reviewer. The only objective assessment in this article is surface electromyography. We apologize for this oversight.

The Rolf Method of Structural Integration is a method of working with the body aimed at restoring the balance of tensions within the myofascial structures. This "restoration" of myofascial tension using the mobilization/manipulation specified by Ida Rolf is to facilitate the central axis of the body to interact with its gravitational vertical, triggering a functional anti-gravity reflex (Jacobson 2011). So-called deep sessions are related to the core muscles, which also include the pelvic floor muscles. The Rolf Method of Structural Integration does not perform an assessment of the patient's body posture, it is rather a tool/technique of working with the body used by many therapists/physiotherapists and sometimes osteopaths.

  1. Reviewer comment: In relation to the results and discussion: It is necessary to measure in the medium term at least to be able to check if the changes are maintained.

Authors’s response:  Yes, of course. We very much agree with this statement, and that was our intention. However, the project continues during the pandemic and lockdown, during which our participants are off-site and no follow-up was possible. Due to the fact that the intervention used by us in the form of "The Rolf method" is in the evaluation phase in terms of evidence-based practice, conducting a follow-up is a priority for us.

We have added some information about follow up. Please see the changes in Study Limitations in line 305.

  1. Reviewer comment: What clinical application do the results have? Is not clear.  

Authors’s response:  Thank you for your question. We have added a fragment of Clinical Implications in the line 303-309.

What is “already known” in this topic:

® In the case of pelvic floor dysfunction, such as urinary incontinence, the conservative treatment of the first choice is pelvic floor muscle training (Cacciari, Dumoulin, and Hay-Smith 2019). However, some studies indicate that exercising or training the PFMs can only be successful if the patient is able to voluntarily contract and relax her PFMs (Berghmans, Seleme, and Bernards 2020). Some studies suggest that body posture may have an effect on pelvic floor function (specifically voluntary contraction and relaxation of PFMs)(Lee 2018).

® Investigation of spatial and temporal responses of soft tissues to applied mechanical forces (mechanotherapeutic interventions such as, for example, The Rolf Method of Structural Intervention) is a growing field of health sciences (Tadeo et al. 2014). 

® Since the goal of The Rolf Method of Structural Integration is to restore the correct body posture and equalize myofascial tensions (according to the principle of biotensegration), we assumed that the holistic approach of this specific method will somehow change the bioelectric activity (sEMG amplitude recording) of the pelvic floor muscles in healthy women.

What this article adds:

The applied intervention in the form of The Rolf Method influenced the PFM sEMG amplitude twofold:

® After the intervention in the test group reduction in sEMG amplitude was noted in the “pre-baseline rest” and “rest after tonic contraction” measurements in the supine position

® After the intervention in the study group, the sEMG amplitude increased in the “phasic contraction” measurements what may suggest that ten IS sessions contributed to an increase in the recruitment of type II (fast-twitch) fibers, responsible for fast, forceful contractions.

These results allow us to continue research in clinical populations: especially in women with stress urinary incontinence (hypotonic PFMs), and with chronic pelvic pain or vulvodynia (hypertonic PFMs). Based on various reports (Franke and Hoesele 2013; Crowle and Harley 2020), we assume that in the case of possible pelvic floor dysfunctions, the patient's body undergoing conservative treatment should be treated as a whole, i.e. all organs, muscles, and structures of the body should be viewed in the context of their setting. A specific dysfunction (e.g. urinary incontinence) cannot be fully understood or treated without perceiving the patient's body as a whole (Findley 2011; Horton 2015).

Berghmans, B., M. R. Seleme, and A. T. M. Bernards. 2020. ‘Physiotherapy Assessment for Female Urinary Incontinence’. International Urogynecology Journal 31 (5): 917–31. https://doi.org/10.1007/s00192-020-04251-2.

Cacciari, Licia P., Chantale Dumoulin, and E. Jean Hay-Smith. 2019. ‘Pelvic Floor Muscle Training versus No Treatment, or Inactive Control Treatments, for Urinary Incontinence in Women: A Cochrane Systematic Review Abridged Republication’. Brazilian Journal of Physical Therapy 23 (2): 93–107. https://doi.org/10.1016/j.bjpt.2019.01.002.

Crowle, Anna, and Clare Harley. 2020. ‘Development of a Biotensegrity Focused Therapy for the Treatment of Pelvic Organ Prolapse: A Retrospective Case Series’. Journal of Bodywork and Movement Therapies 24 (1): 115–25. https://doi.org/10.1016/j.jbmt.2019.10.008.

Findley, Thomas W. 2011. ‘Fascia Research from a Clinician/Scientist’s Perspective’. International Journal of Therapeutic Massage & Bodywork 4 (4): 1–6.

Franke, Helge, and Klaus Hoesele. 2013. ‘Osteopathic Manipulative Treatment (OMT) for Lower Urinary Tract Symptoms (LUTS) in Women’. Journal of Bodywork and Movement Therapies 17 (1): 11–18. https://doi.org/10.1016/j.jbmt.2012.05.001.

Horton, R C. 2015. ‘The Anatomy, Biological Plausibility and Efficacy of Visceral Mobilization in the Treatment of Pelvic FLoor Dysfunction’. J Pelvic Obstet Gynaecol Physiother., no. 117: 5–18.

Jacobson, Eric. 2011. ‘Structural Integration, an Alternative Method of Manual Therapy and Sensorimotor Education’. Journal of Alternative and Complementary Medicine 17 (10): 891–99. https://doi.org/10.1089/acm.2010.0258.

Lee, Kyeongjin. 2018. ‘Activation of Pelvic Floor Muscle During Ankle Posture Change on the Basis of a Three-Dimensional Motion Analysis System’. Medical Science Monitor : International Medical Journal of Experimental and Clinical Research 24 (October): 7223–30. https://doi.org/10.12659/MSM.912689.

Oleksy, Łukasz, Małgorzata Wojciechowska, Anna Mika, Elżbieta Antos, Dorota Bylina, Renata Kielnar, Błażej Pruszczyński, and Artur Stolarczyk. 2020. ‘Normative Values for Glazer Protocol in the Evaluation of Pelvic Floor Muscle Bioelectrical Activity’. Medicine 99 (5): e19060. https://doi.org/10.1097/MD.0000000000019060.

Tadeo, Irene, Ana P. Berbegall, Luis M. Escudero, Tomás Álvaro, and Rosa Noguera. 2014. ‘Biotensegrity of the Extracellular Matrix: Physiology, Dynamic Mechanical Balance, and Implications in Oncology and Mechanotherapy’. Frontiers in Oncology 4 (March). https://doi.org/10.3389/fonc.2014.00039.

Reviewer 2 Report

Dr. Keren Grinberg- Review

This interesting randomized, interventional study "The Rolf Method of Structural Integration and pelvic 2 floor muscle facilitation: a randomized, 3 interventional study "evaluated the Pelvic Floor Muscles (PFMs) after the tenth session of SI by using surface electromyography (sEMG). It seems that the ten SI therapy sessions contributed to the observed changes in the bioelectric activity of the pelvic floor muscles.

SI intervention significantly changes some functional bioelectrical activity of PFMs, providing a basis for further research on the new approach to PFMs facilitation.

Several requests for clarification should be considered:

Line 50: The authors wrote: "Ten sessions, during which the SI 50 therapist changes the position of individual parts of the body in"…..

The frequency of the meetings was once a week but, how long was every session?

Line 53: Can the authors demonstrate the "specific course"? Is it performed in table 1?

Please clarify, how did you assessed the contractions ability?

A number of general comments:

Did the authors address any psychological characteristics that may affect the pelvic floor contractions? These components have also an effect on the contraction of the pelvic floor.

Did the subjects report any pain levels during treatment? Were pelvic floor pain levels measured?

It was interesting to see the urgency and frequency of urination before and after treatment.

Author Response

Reviewer 2

Authors’s response: Thank you very much for your review. We answered point by point on your questions and suggestions:

  1. Reviewer comment: Line 50: The authors wrote: "Ten sessions, during which the SI therapist changes the position of individual parts of the body in”

Authors’s response:  Thank you for this suggestion. We add this information in line 50-53: The Rolf Method of Structural Integration consists of ten sessions during which the therapist changes the position of individual parts of the body concerning the vertical and gravity field to improve the ergonomics of movements and body posture, and possibly alleviate the reported ailments.

  1. Reviewer comment: The frequency of the meetings was once a week but, how long was every session?

Authors’s response: Each intervention lasted for 60 minutes (line 101), we also added this information in line: 86-87.

  1. Reviewer comment: Can the authors demonstrate the "specific course"? Is it performed in table 1?

Authors’s response: Yes, indeed. Each session is performed in Table 1. We added this information in line 53-54: Each session has its specific course, which typically includes manipulation of anatomical segments and all major joints (Jacobson, 2011a). The aims of each session are presented in Table 1.

  1. Reviewer comment: Please clarify, how did you assessed the contractions ability?

Authors’s response: The recording of the sEMG signal occurred when the researcher observed the patient’s ability to perform the contraction and relaxation without any form of help (without any forms of biofeedback). 

The pattern of the pelvic floor muscle activation must be "normal" whereas little data exists on normative sEMG muscle bioelectrical activity (Bo et al., 2017; Oleksy et al., 2020). We recorded sEMG signals when:

® Evaluated sEMG amplitude wasn’t too increased, decreased or absent (elevated amplitude during resting/baseline activity implies that a muscle has insufficient possibilities for rest)

® The timing patterns of muscle bioelectrical activity weren’t too early, late or asynchronous (Oleksy et al., 2020)

® We did not observe the simultaneous bioelectrical activity of the PFM’s synergistic muscles such as: rectus abdominis, gluteus maximus and adductors to make sure that PFMs contractions were isolated. ®

® Moreover, during the contraction, each participant should not stop breathing, tilt the pelvis, or strain (Neels et al., 2018)

We added an extract from the above information to Table 2. Please see the line: 122-123

  1. Reviewer comment: Did the authors address any psychological characteristics that may affect the pelvic floor contractions? These components have also an effect on the contraction of the pelvic floor.

Authors’s response: Thank you very much for this question. Currently, we are working with Psychological Questionnaires with the next group of the project participants: Subjective evaluation of own mental health, Locus of Control (internal and external) and Social Desirability, Selected parameters of Body Image and Emotional Intelligence - Emotional Intelligence Questionnaire.

  1. Reviewer comment: Did the subjects report any pain levels during treatment? Were pelvic floor pain levels measured?

Authors’s response: The subjects did not report any pain during the intervention and pelvic floor pain was not measured. We consider using Pain Scales and vaginal palpation in aim to diagnose the PFM trigger points in women with chronic pelvic pain or vulvodynia. But this is a plan for our future work.

Reviewer comment: It was interesting to see the urgency and frequency of urination before and after treatment.

Authors’s response: Yes, indeed. In our project, we want to work with pelvic floor dysfunction in women of all ages and we are going to use for example Pad-Test, voiding diary, or ICIQ-SF. Thank you very much for your comment.

Bo, K. et al. (2017) ‘An International Urogynecological Association (IUGA)/International Continence Society (ICS) joint report on the terminology for the conservative and nonpharmacological management of female pelvic floor dysfunction’, Neurourology and Urodynamics, 36(2), pp. 221–244. doi: 10.1002/nau.23107.

Neels, H. et al. (2018) ‘Common errors made in attempt to contract the pelvic floor muscles in women early after delivery: A prospective observational study’, European Journal of Obstetrics, Gynecology, and Reproductive Biology, 220, pp. 113–117. doi: 10.1016/j.ejogrb.2017.11.019.

Oleksy, Ł. et al. (2020) ‘Normative values for Glazer Protocol in the evaluation of pelvic floor muscle bioelectrical activity’, Medicine, 99(5), p. e19060. doi: 10.1097/MD.0000000000019060.

Round 2

Reviewer 1 Report

THANKS FOR THE CLARIFICATIONS IN THE TEXT. IF THE STUDY CONTINUES TO GO, IT WOULD BE BETTER TO PUBLISH FINAL RESULTS. THEY CAN HELP DEFINE FUTURE INTERVENTIONS.